# UDON: Universal Dynamic Online distillatioN for generic image representations

**Nikolaos-Antonios Ypsilantis**[*1] **Kaifeng Chen**[2] **André Araujo**[2] **Ondřej Chum**[1]

[1]VRG, FEE, Czech Technical University in Prague   [2]Google DeepMind

## Abstract

Universal image representations are critical in enabling real-world fine-grained and instance-level recognition applications, where objects and entities from any domain must be identified at large scale. Despite recent advances, existing methods fail to capture important domain-specific knowledge, while also ignoring differences in data distribution across different domains. This leads to a large performance gap between efficient universal solutions and expensive approaches utilising a collection of specialist models, one for each domain. In this work, we make significant strides towards closing this gap, by introducing a new learning technique, dubbed UDON (Universal Dynamic Online distillatioN). UDON employs multi-teacher distillation, where each teacher is specialized in one domain, to transfer detailed domain-specific knowledge into the student universal embedding. UDON's distillation approach is not only effective, but also very efficient, by sharing most model parameters between the student and all teachers, where all models are jointly trained in an online manner. UDON also comprises a sampling technique which adapts the training process to dynamically allocate batches to domains which are learned slower and require more frequent processing. This boosts significantly the learning of complex domains which are characterised by a large number of classes and long-tail distributions. With comprehensive experiments, we validate each component of UDON, and showcase significant improvements over the state of the art in the recent UnED benchmark. Code: `https://github.com/nikosips/UDON`.

## 1 Introduction

Imagine you point your cellphone at anything, and it tells you what it is, be it tangerine chicken with rice, Mk1 Volkswagen Rabbit Cabriolet, statue of Aquaman, Pasadena City Hall, or Yorkshire Terrier. Such a product is the ultimate goal of fine-grained and instance-level visual recognition. The key component enabling such an application is a general-purpose image representation, or equivalently image embedding, designed to handle imagery of varied domains at scale. Traditionally, image embedding models have been developed for specific domains separately [29, 34, 17, 9], such as landmarks [32], products [41], clothes [22], faces [40], to name just a few. However, as visual recognition applications grow in popularity and scope [47, 1, 2], it is impractical to handle images of different object types with specialized, per-domain models. A potential solution for this problem is to leverage recent foundation models, such as CLIP [33] or DINOv2 [26], which have been proposed to enable a wide variety of multimodal applications. Even though these models possess a broad visual understanding, they tend to lack detailed fine-grained knowledge off-the-shelf [45], which is critical in practice. For this reason, recent efforts aim at developing universal embedding solutions that can generalize to handle multiple fine-grained object types with a single model [45, 39], ensuring scalability in real-world scenarios.

---

[*]Corresponding author: `ypsilnik@fel.cvut.cz`

38th Conference on Neural Information Processing Systems (NeurIPS 2024).

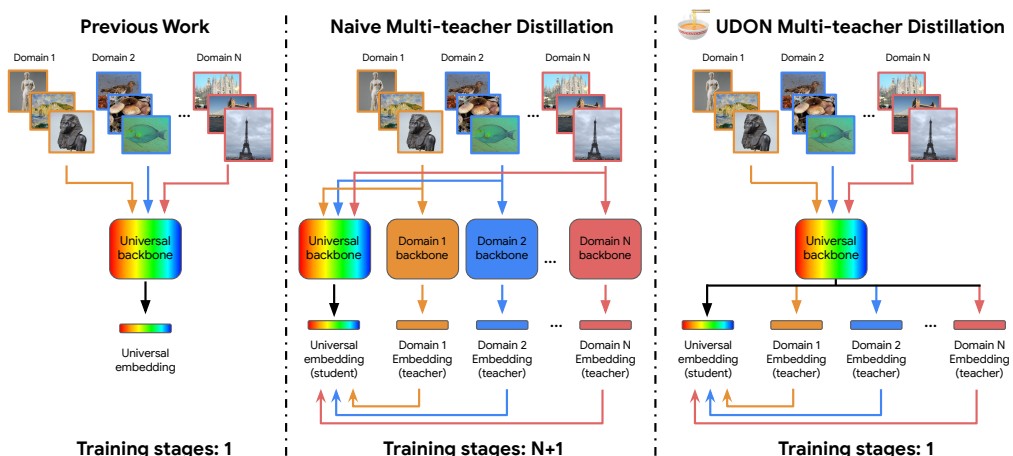

Figure 1: Training of a universal embedding on multiple fine-grained visual domains. The baseline approach of [45] (left) uses classification loss across training classes from all domains. It is prone to cancelling out contradicting cues from different domains. To overcome this issue, a naive multi-teacher distillation approach (middle) first trains one specialized teacher per domain (with a classification loss) to capture domain specifics, then distils them to the universal embedding (student). Our proposed Universal Dynamic Online distillatioN – UDON (right) jointly trains the specialized teacher embeddings and the universal embedding (student) with classification and, at the same time, distils the teacher embeddings to the universal embedding. Due to joint training of a shared backbone, UDON scales to a large number of domains.

There are two main challenges in training such universal models addressed in the paper. First, it is difficult to encode detailed knowledge about many image domains in a single model. In several cases, features that are helpful for one object type may be useless for others. One example is the importance of color: while for food recognition it is critical to discriminate red curry from green curry, when recognizing car models both a red and a green Toyota Corolla LE 2024 would belong to the same class. This makes the training of universal models from data of different domains particularly hard, since the data may present conflicting peculiarities across domains, leading to sub-optimal learning. To address this issue, we propose a novel knowledge distillation approach, where one teacher model is trained for each domain individually, and a student universal embedding model learns from the collection of teachers. This setup allows the specialized teachers to capture domain-specific knowledge, which is then transferred into the universal embedding. However, if this distillation setup is deployed naively, one may incur substantial costs, as a separate teacher model would need to be trained for each domain. For this reason, we propose to share the backbone between all teachers and the universal student, reaching a solution that achieves high performance and incurs small additional training costs, as all models are jointly trained in an online manner – see Figure 1.

The second main challenge we highlight is that different domains may present data with vastly different distributions: *e.g.*, while one domain may present a moderate number of classes with a roughly balanced number of samples, in others, the number of classes may be very large, and the distribution of samples long-tailed. This means that different domains may require different training curricula for a model to properly learn their characteristics. This leads us to propose a sampling technique that dynamically selects which domains will be processed at each point in the learning process based on the losses measured on the fly. With this method, we demonstrate significant improvements to the performance of the most challenging domains, which are learned slower and require more frequent processing compared to other domains.

**Contributions.** To summarize, in this work we introduce the following contributions. **(1)** We leverage knowledge distillation to infuse the **universal** embedding model with the learnings of specialist, per-domain teacher models. In our novel training setup, the model backbone for all teacher embeddings and the student universal embedding is shared, leading to an efficient method where all models are jointly trained in an **online** manner. Our findings show that sharing the backbone facilitates the distillation process significantly, even reaching performance that surpasses the distillation from

separate specialist teachers. **(2)** To enable an appropriate training regime in a universal embedding setup, we propose to adapt the learning process **dynamically**, adjusting the sampling of image domains based on their losses, which appears suitable to visual knowledge spanning a range of fine-grained domains. We show that this can help substantially with more complex domains that require more frequent model updates to enhance their performance. **(3)** We perform comprehensive experiments on the recent UnED [45] dataset, that highlight the value of each proposed component, as well as compare our technique against competitor approaches. Our complete method, named **UDON** (**U**niversal **D**ynamic **O**nline distillatio**N**), showcases state-of-the-art results that boost Recall@1 by up to 2.3%.

## 2 Related Work

**Knowledge distillation (KD).** Initially proposed to transfer the knowledge of large and complex models to smaller and faster ones [14], standard KD trains a light student to mimic the softmax outputs produced by a heavy teacher. Tailoring KD to representation learning, [27, 28] distill relations between image representations and [19] focus on retrieval rankings. Online KD [12] lifts the need for separate two-stage training for the teacher and the student and trains them simultaneously. Multi-teacher KD [20, 15] aims to transfer the knowledge of multiple teachers into one student model. [24] performs multi-teacher KD for a single domain visual retrieval, by aggregating the teacher relations in a single target that the student should mimic. Differently than [15], which proposes an online multi-teacher distillation approach that trains a different backbone for each teacher, we propose to share a common backbone between all the teachers and the student, showing improved performance while also being much more efficient. [50] combines online and multi-teacher knowledge distillation in a single multi-branch network, training an ensemble of teachers on the fly. Differently from [50, 24], each of the teachers in our KD approach is specialized to only a fraction of the data (a single domain), being relevant only to part of the universal embedding task. Similarly to [50], we also create an online multi-branch architecture, however the teachers are not updated by the other teachers' knowledge, as they are related to different visual domains. Additionally, our teachers transfer relational knowledge to the student, since our focus is on learning image embeddings, in contrast to [50] which only focuses on classification.

**Universal representation learning.** Learning a representation that generalizes and can be reused efficiently across visual domains is a long-standing goal in computer vision. [4] introduces domain-specific normalization to make classification networks generalize to multiple visual domains, while [35] introduces adapter modules to improve the accuracy of domain-specific representations. [23] proposes a multi-task vision and language universal representation trained on 12 different datasets. However, this body of work assumes knowledge of the test time domain, which does not hold in our setup. Recent large visual foundational models [26, 33] that are trained on large amounts of data with diverse objectives show great zero-shot performance on a number of downstream visual tasks, making them great candidates for universal embedding applications. However, [45] shows that these models, even though generalizing to many diverse domains, cannot effectively handle instance-level and fine-grained domains (which are the focus of this work) without further fine-tuning. [37] shows that appending an MLP projector between the objective and the representation used for the downstream tasks improves the generalisability of the representation, inspiring a multi-domain variant we use as a baseline in this work. In [3], a multi-domain representation for fine-grained retrieval is learned, which utilizes no labels for training. Differently from it, we focus on the supervised task setup of [45], which constitutes a much larger-scale problem that additionally includes instance-level domains. [20] introduces distillation as a way to learn universal representations, while [10] tailors multi-teacher distillation to universal embedding learning. Differently from [20, 10], we do not use task-specific backbones that are costly to scale across a large number of domains. While [10] tackles a universal embedding setup, the effectiveness of their method is only assessed on a small dataset, where three small domains at a time are distilled into a universal representation. In contrast, we tackle learning in a more practical large-scale setup, with an efficient approach that distills knowledge from eight diverse visual domains into the universal embedding. Recently, the UnED dataset was introduced in [45] as a new large-scale benchmark for universal embeddings. Their experiments considered the training of models only via classification objective, with different sampling and classifier configurations. Setting our approach apart is that we go beyond to capture detailed knowledge from diverse domains via distillation, besides proposing a more suitable training dynamic that can accommodate data from diverse domains. Concurrently, UNIC [38] proposes a universal classification model using

multi-teacher distillation. While our approach trains a student embedding from multiple teachers specialized in fine-grained visual domains, UNIC distills foundational models trained for diverse tasks, such as semantic segmentation and classification, with both supervised and self-supervised objectives.

**Dynamic sampling.** When training in a multi-task setting, the sampling frequency of the different domains can greatly affect final performance. Poly-ViT [21] explored different samplings tailored to their multi-modal multi-task model, and concluded that sampling each domain with a weight proportional to the number of training steps that a corresponding specialized model needs to achieve maximum performance works the best, while [45] additionally comes to the same conclusion for the task of universal image embedding. This approach is costly with an increasing number of domains in hand, as one model for each domain needs to be trained, which inspires us to discover more efficient sampling strategies. [23] proposes Dynamic Stop-and-Go sampling, which updates the sampling weight of each domain based on the validation set accuracy. Differently from them, we propose to calculate the sampling weights only based on training loss, which doesn't require the expensive feature extraction of the validation set but can happen on the fly. A similar idea has been explored by [31] in the context of pre-training vision-language models, which is far from this work's, as we focus on learning multi-domain fine-grained image embeddings.

## 3 Proposed method

This section presents our proposed training method, Universal Dynamic Online distillatioN (UDON), to learn the universal image embedding. UDON utilizes a pretrained Vision Transformer [8] as the image encoder, which is further fine-tuned with a combination of classification and distillation objectives. First, some preliminaries concerning the backbone architecture that we build upon are introduced, and afterward, the complete training pipeline is presented in detail.

### 3.1 Preliminaries

The Vision Transformer [8] backbone produces the [CLS] token as a global representation of the image. Let $e_b : \mathcal{X} \to \mathbb{R}^D$ denote the Vision Transformer as a function that takes an input image $x \in \mathcal{X}$ and maps it to the [CLS] token $e_b(x) \in \mathbb{R}^D$, compactly denoted as $e_b$. The dimensionality $D$ is backbone dependent and usually higher than the one required in the downstream task, hence projection to lower dimensional space is introduced. The final vector after projection is the universal embedding, denoted as $e_u \in \mathbb{R}^d$, $d < D$. Following standard practice used in image retrieval architectures [11], $e_b$ and $e_u$ are $\ell_2$ normalized. When referring to a batch of embeddings, we use capitalized notation, *e.g.*, a batch of embeddings $e_u$ is denoted $E_u \in \mathbb{R}^{d \times B}$, where $B$ is the batch size. For training with a classification loss on top of the universal embedding in the multi-domain setup, a Separate Classifier (SC) per domain is employed, classifying across the classes of that specific domain, an option justified by [45]. In the following, the word "head" denotes both the projection to the embedding space and the classifier to which the projected embedding is input.

### 3.2 Universal Dynamic Online distillatioN (UDON)

Our UDON training approach introduces an efficient multi-teacher distillation method, relying on a shared feature extraction backbone. The entire training pipeline is presented in Figure 2. The backbone produces the initial high dimensional image embedding, $e_b$, for the samples of all domains. Given $e_b$, in addition to projecting it into the universal embedding space $e_u$ that is used at test time (as in [45]), we also project $e_b$ to per-domain spaces, which constitute the teacher embeddings $\{e_{t_i} \in \mathbb{R}^{D_t}\}_i$. Teacher $i$ is only activated for samples of domain $i$. Both the universal and the domain-specific (teacher) projections are realized by linear layers, and there is a domain-specific projection for each domain.

The universal embedding is trained with both classification and distillation objectives, calculated on batches containing a single domain at a time. While training with a classification objective allows grasping broad knowledge for several domains, it may lead to a sub-optimal model due to contradictory cues when combining data from all domains. Therefore, we employ distillation from the domain-specific teachers to infuse the universal model with domain-specific knowledge. The loss functions are presented below.

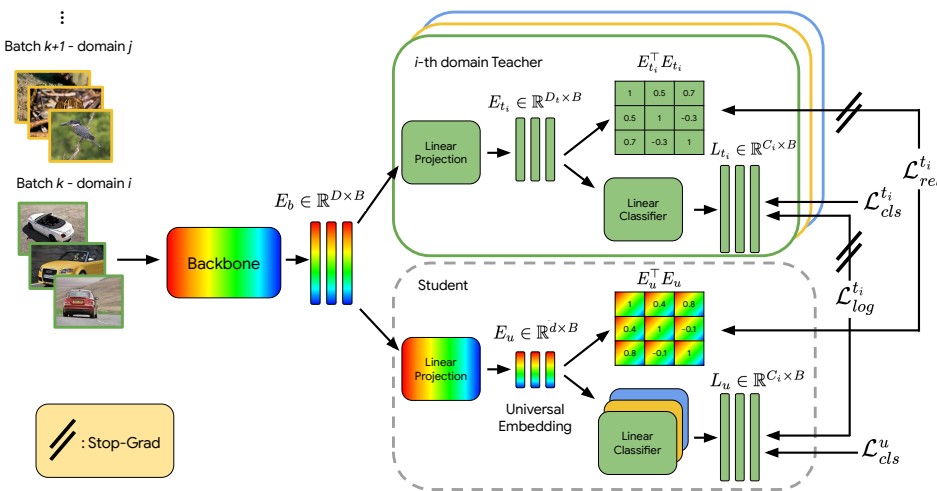

Figure 2: **Block diagram of UDON's training process**. Each batch of size $B$ contains images from a single domain (*e.g.*, cars, natural world, etc). When a batch with domain $i$ is processed, the $i$-th teacher head is used. Both the teacher and the student employ a classification loss ($\mathcal{L}_{cls}^{t_i}$, $\mathcal{L}_{cls}^u$) on top of their batched logits ($L_{t_i}$, $L_u$), predicting among $C_i$ classes. The student is additionally trained via distillation, by learning intra-batch relationships ($\mathcal{L}_{rel}^{t_i}$) and logits ($\mathcal{L}_{log}^{t_i}$) with the domain teacher guidance. Note that the distillation losses are backpropagated only through the student's head.

**Classification losses.** The universal and the domain-specific embeddings are trained with classification losses (Normalized Softmax Loss [48]), instantiated with separate classifiers for each domain. These losses update the backbone via gradients propagated through the teacher and the student heads, while teacher-specific or student-specific parameters are only updated via gradients from their respective heads. The $i$-th teacher (of domain $i$) classification loss $\mathcal{L}_{cls}^{t_i}$ and the universal embedding (student) classification loss $\mathcal{L}_{cls}^u$ are defined as:

$$\mathcal{L}_{cls}^{t_i} = -\frac{1}{B}\sum_{j=1}^{B} y_j \log(\hat{y}_j^{t_i}) \qquad \text{and} \qquad \mathcal{L}_{cls}^u = -\frac{1}{B}\sum_{j=1}^{B} y_j \log(\hat{y}_j^u), \qquad (1)$$

where $B$ is the batch size, $y_j$ is the one-hot ground truth vector of sample $j$, $\hat{y}_j^{t_i}$ is the predicted probability vector for sample $j$ produced from teacher of domain $i$ and $\hat{y}_j^u$ is the predicted probability vector for sample $j$ produced from the universal embedding (student). It is important to note that the classifiers of the student and the teachers are different and that the student employs as many classifiers as the number of domains (SC).

**Distillation losses.** The student is tasked to match the teacher embedding of the corresponding domain by enforcing two separate distillation losses. The first is a relational distillation loss, which acts on batch similarity matrices. Given the student's batch embeddings $E_u \in \mathbb{R}^{d \times B}$, its batch similarity matrix is formed as $E_u^\top E_u \in \mathbb{R}^{B \times B}$. Similarly, for the $i$-th teacher's batch embeddings $E_{t_i} \in \mathbb{R}^{D_t \times B}$, its batch similarity matrix is formed as $E_{t_i}^\top E_{t_i} \in \mathbb{R}^{B \times B}$. The goal is for the student to learn detailed intra-domain similarities from the more powerful domain-specific teacher. Specifically, the student's intra-domain cosine similarities are encouraged to follow the $i$-th teacher's cosine similarities, when the batch of images comes from domain $i$:

$$\mathcal{L}_{rel}^{t_i} = ||E_u^\top E_u - E_{t_i}^\top E_{t_i}||^2. \qquad (2)$$

Additionally, the student is tasked to match its logits to the teacher's, minimizing their KL divergence, after scaling with identical temperature $T$ and softmax normalization of both:

$$\mathcal{L}_{log}^{t_i} = \text{KL}\left(\text{softmax}(\boldsymbol{l}_u/T) \; || \; \text{softmax}(\boldsymbol{l}_{t_i}/T)\right), \qquad (3)$$

where $\boldsymbol{l}_u$ is the logit vector produced by the classifier on the student's side and $\boldsymbol{l}_{t_i}$ is the logit vector produced by the classifier on the $i$-th teacher's side. The temperature $T$ is shared across all teachers

and the student. This loss provides a more global context, as it captures the similarities between an embedding and all of the class prototypes in the domain (which exist in the classifier), instead of only relating embeddings in a batch. Both distillation losses do not backpropagate through the domain-specific teacher head, as only the student should try to learn from the teacher. Distillation starts at the beginning of the training and it happens in an online manner, at the same time that the universal student and the teachers are trained with the classification losses. The total loss for a training batch, containing images of domain $i$, is as follows:

$$\mathcal{L}_{total} = \mathcal{L}_{cls}^{t_i} + \mathcal{L}_{cls}^{u} + \mathcal{L}_{rel}^{t_i} + \mathcal{L}_{log}^{t_i}. \tag{4}$$

**Dynamic domain sampling.** Each domain comes with its own training data, which differ in the number of classes and in the number of examples. Balancing of the domains is performed through sampling of the training data. In [45], training is performed with clean batches, *i.e.* each batch only contains examples from a single domain. Three different sampling schemes were compared in [45]. In Dataset Size sampling, the datasets are sampled proportionally to their size, biasing towards large datasets. In Round-Robin (RR) sampling, the domains are sampled equally often in a cyclic order. The Specialist Steps sampling selects the domains proportionally to the number of steps the specialist at the particular domain needs to achieve maximum validation performance. Only the last approach takes into account the difficulty of the classification task in each domain. However, such sampling is static, does not reflect possible correlations in similar domains, and requires the training of a specialist for each domain prior to training the universal model.

We propose to sample each domain proportionally to the classification loss produced by the domain-specific embedding during training, relative to the corresponding loss of the other domain-specific embeddings. This way, the domains for which the training loss decreases slower than for other domains are sampled more for training. More specifically, we sample the domain of the next training batch out of the distribution which for each domain $m$ assigns the probability:

$$P(m) = \frac{\text{train loss}_m}{\sum_{i=1}^{N} \text{train loss}_i}, \tag{5}$$

where $N$ is the number of domains, train loss$_m$ is the classification loss produced by the embedding of domain $m$, and the denominator represents the sum of the corresponding losses across all domains. This distribution is updated after every $S$ steps, a hyperparameter that is tuned on the validation set.

## 4 Experiments

### 4.1 Experimental settings

**Dataset.** The proposed method is evaluated on the recent Universal Embeddings Dataset (UnED) [45], the largest dataset for multi-domain fine-grained retrieval. It comprises $4.1$M images, with $349$k classes distributed across $8$ image domains: food (Food2k dataset) [25], cars (CARS196 dataset) [18], online products (SOP dataset) [41], clothing (InShop dataset) [22], natural world (iNat dataset)[43], artworks (Met dataset) [46], landmarks (GLDv2 dataset) [44] and retail products (Rp2k dataset) [30]. We follow the train-validation-test splits and the evaluation protocol defined in [45], a brief review follows. Each index image and each query in the test set are described by a 64-D (universal) embedding, and Euclidean nearest neighbors in the embedding space are found for each query among the index set. The index contains images from all $8$ domains combined; hence, cross-domain false positives are possible. Performance is measured by two metrics: Recall@1 (R@1), which is equivalent to the correctness of the top neighbor, and modified Mean Precision@5 (P@5) [45]. The average performance over the queries in each domain is reported, as well as the balanced average of these values across all domains.

**Compared methods.** The universal embedding task is relatively new and only a few baselines were published in [45, 6]. We extend these baselines by re-purposing the single-domain embedding method of [37] (Table 1). Additionally, we also compare to two straight-forward multi-domain distillation methods [10] (Table 3). The main baselines compared with this work are the best-performing methods from [45], namely USCRR, UJCRR, and USCSS, which vary the classifier setup (SC: Separate Classifier, JC: Joint Classifier) and the sampling (RR: Round Robin, SS: Specialist Steps). We also evaluate a variant of USCRR, which uses the proposed dynamic sampling scheme, dubbed "USC w/ Dyn Sampler". The USCRR [45] method (USC w/ Dyn Sampler) is further expanded by

| Model | Food2k | | CARS196 | | SOP | | InShop | | iNat | | Met | | GLDv2 | | Rp2k | | Mean | |
|---|---|---|---|---|---|---|---|---|---|---|---|---|---|---|---|---|---|---|
| | P@5 | R@1 | P@5 | R@1 | P@5 | R@1 | P@5 | R@1 | P@5 | R@1 | P@5 | R@1 | P@5 | R@1 | P@5 | R@1 | P@5 | R@1 |
| **Off-the-shelf** | | | | | | | | | | | | | | | | | | |
| IN [42](**768-D**) | 31.1 | 44.1 | 41.4 | 54.1 | 43.7 | 65.6 | 35.5 | 53.9 | 67.1 | 74.2 | 21.1 | 30.8 | 14.8 | 25.2 | 52.9 | 74.3 | 38.4 | 52.8 |
| IN + MIM [6](**768-D**) | – | 36.6 | – | 52.3 | – | 53.2 | – | 40.2 | – | 68.2 | – | 20.2 | – | 17.8 | – | 60.7 | – | 43.7 |
| CLIP [33](**768-D**) | 29.4 | 42.9 | 74.7 | 82.2 | 44.2 | 65.4 | 37.2 | 56.0 | 52.4 | 61.9 | 21.4 | 28.5 | 20.4 | 31.0 | 38.6 | 59.9 | 39.8 | 53.5 |
| DINOv2 [26](**768-D**) | 39.9 | 51.4 | 67.1 | 79.5 | 35.6 | 56.0 | 17.4 | 33.4 | 71.2 | 77.6 | 38.3 | 48.1 | 35.4 | 51.7 | 46.6 | 67.8 | 43.9 | 58.2 |
| SigLIP [49](**768-D**) | 39.5 | 52.8 | 93.9 | 95.7 | 50.8 | 69.7 | 53.5 | 73.1 | 59.6 | 67.5 | 31.6 | 41.2 | 20.6 | 32.0 | 42.7 | 64.3 | 49.0 | 62.0 |
| **Specialist+Oracle** | | | | | | | | | | | | | | | | | | |
| IN Specialist+Oracle | 49.9 | 62.8 | 61.9 | 71.8 | 60.9 | 78.1 | 66.3 | 85.9 | 70.1 | 75.2 | 20.4 | 24.9 | 31.2 | 43.1 | 73.6 | 87.1 | 54.9 | 66.6 |
| CLIP Specialist+Oracle | 51.5 | 63.7 | 83.4 | 88.5 | 65.8 | 81.2 | 68.0 | 86.2 | 67.3 | 73.0 | 27.6 | 32.9 | 35.1 | 46.6 | 69.7 | 84.4 | 59.6 | 70.4 |
| **ImageNet21k pretraining** | | | | | | | | | | | | | | | | | | |
| [45] UJCRR | 48.6 | 60.3 | **62.9** | **71.3** | **64.7** | **80.2** | **74.0** | **89.9** | 68.3 | 73.3 | 5.5 | 7.0 | 21.1 | 31.6 | 74.1 | 86.8 | 52.4 | 62.6 |
| [45] USCRR | 48.3 | 60.9 | 58.9 | 69.7 | 61.9 | 78.7 | 70.4 | 88.3 | 69.1 | 74.2 | 7.3 | 9.7 | 21.3 | 31.4 | 74.1 | 87.1 | 51.4 | 62.5 |
| [45] USCSS | 49.0 | 61.7 | 53.4 | 64.3 | 62.0 | 78.8 | 67.6 | 87.2 | 68.3 | 73.5 | 8.4 | 10.7 | 28.0 | 40.6 | 73.5 | 87.1 | 51.3 | 63.0 |
| USC w/ Dyn Sampler | 46.2 | 59.1 | 56.3 | 67.4 | 61.1 | 78.4 | 65.9 | 86.1 | 69.6 | 74.8 | 12.0 | 15.6 | 25.9 | 37.3 | 73.2 | 86.9 | 51.3 | 63.2 |
| MLP baseline w/ Dyn Sampler | 47.5 | 60.1 | 51.3 | 63.0 | 61.8 | 78.5 | 64.1 | 85.0 | 69.7 | 74.8 | **14.3** | **17.8** | 25.8 | 36.4 | 73.3 | 86.8 | 51.0 | 62.8 |
| **UDON (Ours)** | **49.6** | **62.2** | 61.3 | 71.2 | 64.2 | **80.2** | 69.8 | 88.5 | **70.4** | **75.3** | 12.0 | 15.9 | **28.6** | **40.9** | **75.6** | **88.0** | **53.9** | **65.3** |
| **CLIP pretraining** | | | | | | | | | | | | | | | | | | |
| [45] UJCRR | 50.1 | 62.0 | **80.0** | 85.4 | **68.6** | **82.7** | **77.0** | **91.1** | 63.7 | 69.5 | 4.6 | 5.8 | 25.5 | 36.0 | 70.1 | 84.1 | 55.0 | 64.6 |
| [45] USCRR | 49.5 | 61.4 | 79.0 | 84.9 | 65.6 | 81.1 | 73.1 | 89.4 | 64.4 | 70.5 | 8.6 | 10.8 | 25.3 | 36.5 | 71.1 | 85.1 | 54.6 | 64.9 |
| [45] USCSS | 49.8 | 62.0 | 76.4 | 83.4 | 65.8 | 81.3 | 71.0 | 88.5 | 65.3 | 71.4 | 9.9 | 12.7 | **31.5** | 42.8 | 70.1 | 84.8 | 55.0 | 65.9 |
| **UDON (Ours)** | **50.3** | **62.4** | **80.0** | **85.8** | 67.0 | 82.1 | 71.8 | 89.7 | **66.7** | **72.7** | **15.8** | **19.6** | 30.9 | **43.4** | **72.7** | **85.9** | **56.9** | **67.7** |

Table 1: Performance comparison of the universal embedding for the proposed UDON method against the previous state-of-the-art and the proposed baselines, on the test set of UnED dataset. Off-the-shelf models are shown for reference, as they employ much higher dimensional descriptors (768-D vs. 64-D) than the rest of the methods. For each type of pre-training, the best method is highlighted in bold. The Specialist+Oracle model constitutes a non-realistic method that is presented in order to get an estimate of the maximum performance that can be achieved in each domain. All of the methods use the ViT-Base/16 backbone.

inserting an MLP projector [37] between the universal embedding and its a classifier for each domain. The projectors consist of three hidden layers of sizes 256, 256, and 512 respectively. This new baseline method is referred to as the "MLP baseline". We compare with the off-the-shelf embeddings from ImageNet21k (IN) [42], finetuned ImageNet21k model with masked image modeling (IN+MIM) [6], CLIP [33], DINOv2 [26], and SigLIP [49], which utilize embeddings of much higher dimensionality (768D vs 64D). Lastly, we compare against the Specialist+Oracle baseline, a non-realistic model proposed in [45] to get a hypothetical estimate of the maximum performance that can be achieved on each individual domain, by choosing the specialist of the query's domain as the embedding for both the query and the index set. All of the methods (including UDON), use the ViT-Base/16 backbone, and additionally, apart from the off-the-shelf ones, are fine-tuned on the training set of UnED.

**Implementation details.** For fair comparisons with the baselines of [45], identical values for common hyperparameters are used. The newly introduced hyperparameters are tuned based on performance on the validation set of UnED. For the KL divergence loss (3), the value of temperature $T$ is set to $T = 0.1$ (a discussion regarding this choice can be found in the Appendix); the teacher embeddings have dimensionality of $D_t = 256$; the four loss components contribute equally to the total loss $\mathcal{L}_{total}$ (no weights need to be tuned). We set the universal student embedding dimensionality to $d = 64$ for direct comparability against previous work. The batch size is set as $B = 128$. The hyperparameter $S$ for the number of steps, after which the dynamic sampler is updated, is set to 1000. Each experiment is repeated 3 times with different seeds; the reported values are averaged over those runs. The standard deviations are reported in the Appendix. The ViT-Base/16 variant of the Vision Transformer is used as the backbone with ImageNet21k [42] and CLIP [33] initializations. The linear projections are initialized randomly, as well as the corresponding classifiers. Our implementation is based on the Scenic framework [7], a library based on Jax [5]/Flax [13]. Experiments are executed on Google Cloud TPU v4s [16].

## 4.2 Main results

In Table 1, we present the performance of UDON and the compared methods. For the full UDON method, we present results for two different pretrainings for more complete comparisons, namely ImageNet21k [42], and CLIP [33]. For the "MLP baseline" and the " [45] USC w/ Dyn Sampler" baseline, we present results for ImageNet21k pretraining only.

On average, as well as on most of the individual domains, the proposed UDON method achieves state-of-the-art performance, for both types of pretraining (ImageNet21k and CLIP). For ImageNet21k, it achieves an improvement of 1.5% and 2.3% on mean P@5 and R@1, respectively, over the previous state-of-the-art. For CLIP, it achieves an improvement of 1.9% and 1.8% on mean P@5 and R@1, respectively, over the previous state-of-the-art. The biggest improvements are observed in the Met,

GLDv2 and iNat domains, all of which are characterized by a large number of classes and long-tail distribution. Our proposed method makes notable progress towards closing the performance gap to the Specialist+Oracle baseline, coming as close as 1%-1.3% for the respective metrics, for ImageNet21k pretraining. The MLP baseline [37] with the addition of dynamic sampling ("MLP baseline w/ Dyn Sampler") shows comparable performance on average with the previously reported state-of-the-art methods of [45] and the baseline method USCRR with Dyn Sampler instead of RR sampling ("USC w/ Dyn Sampler"), showing that it is not trivial to extend the conclusions of [37] to the multi-domain embedding setting. The proposed UDON method achieves an improvement of 2.9% and 2.5% on mean P@5 and R@1, respectively, over the "MLP baseline w/ Dyn Sampler", and 2.6% and 2.1% on mean P@5 and R@1 respectively, over the "USC w/ Dyn Sampler" baseline. We additionally perform an experiment where we combine the MLP baseline [37] with UDON, by appending an MLP projector between the classifier of every domain and the universal embedding. This underperforms the UDON method by 0.6% and 0.3% on mean P@5 and R@1 respectively, while bringing significant extra cost on the number of parameters of the model. In Figure 3 qualitative results for two queries of the UnED test set are shown, for which the UDON universal embedding exhibits better retrieval performance compared to the USCRR [45] baseline embedding.

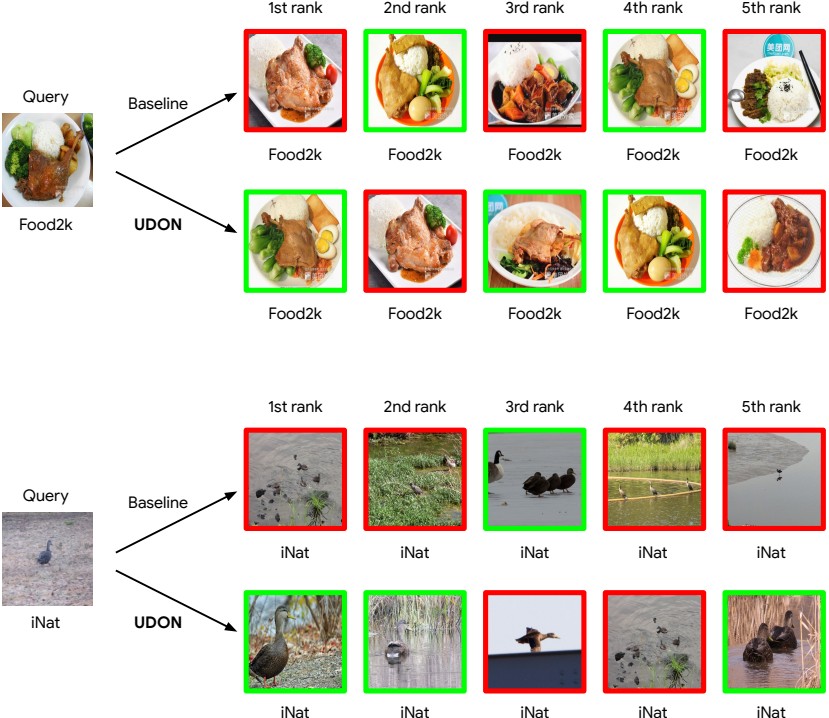

Figure 3: **Qualitative results for our UDON method.** We present the 5 nearest neighbours that are retrieved by the baseline (USCRR) embedding (top row) and the proposed UDON embedding (bottom row), for queries that the proposed method improves over the baseline. Each image shows the domain it comes from (underneath it). The correct neighbors are in green border, the incorrect ones are in red.

### 4.3 Ablations

A spectrum of methods employing various components of UDON are evaluated to examine the impact of each individual block. The methods range from the baseline USCRR (Table 2, row 1) to the full UDON (Table 2, row 3) method. All ablations are initialized by ImageNet21k pretraining. Qualitative results comparing the full UDON method with the baseline "USCRR" are presented in the Appendix.

**Dynamic sampler.** Two methods are evaluated to demonstrate the importance of the dynamic sampler (DS). The baseline with DS (Table 2, row 2 – compare with row 1) and UDON without DS (Table 2,

| Model | Food2k | | CARS196 | | SOP | | InShop | | iNat | | Met | | GLDv2 | | Rp2k | | Mean | |
|---|---|---|---|---|---|---|---|---|---|---|---|---|---|---|---|---|---|---|
| | P@5 | R@1 | P@5 | R@1 | P@5 | R@1 | P@5 | R@1 | P@5 | R@1 | P@5 | R@1 | P@5 | R@1 | P@5 | R@1 | P@5 | R@1 |
| **1** [45] USCRR | 48.3 | 60.9 | 58.9 | 69.7 | 61.9 | 78.7 | 70.4 | 88.3 | 69.1 | 74.2 | 7.3 | 9.7 | 21.3 | 31.4 | 74.1 | 87.1 | 51.4 | 62.5 |
| **2** USC w/ Dyn Sampler | 46.2 | 59.1 | 56.3 | 67.4 | 61.1 | 78.4 | 65.9 | 86.1 | 69.6 | 74.8 | 12.0 | 15.6 | 25.9 | 37.3 | 73.2 | 86.9 | 51.3 | 63.2 |
| **UDON (Ours)** | | | | | | | | | | | | | | | | | | |
| **3** Full UDON | 49.6 | 62.2 | 61.3 | 71.2 | 64.2 | 80.2 | 69.8 | 88.5 | **70.4** | 75.3 | 12.0 | 15.9 | 28.6 | 40.9 | 75.6 | 88.0 | 53.9 | **65.3** |
| **4** w/o Dyn Sampler (RR) | **50.3** | **62.9** | **62.9** | **72.5** | **67.4** | **82.2** | **75.4** | **90.8** | **70.4** | **75.4** | 7.6 | 9.4 | 23.3 | 33.4 | **76.8** | **88.6** | **54.3** | 64.4 |
| **5** 64-D teachers | 47.7 | 60.3 | 60.2 | 70.3 | 64.3 | 80.3 | 68.6 | 87.6 | 69.9 | 75.0 | 11.3 | 14.5 | 26.4 | 38.1 | 74.2 | 87.6 | 52.8 | 64.2 |
| **6** w/o logit distillation | 49.2 | 61.8 | 60.1 | 70.4 | 62.6 | 79.2 | 68.0 | 87.2 | 70.3 | 75.3 | 12.5 | 15.6 | **28.7** | **41.0** | 74.9 | 87.8 | 53.3 | 64.8 |
| **7** w/o any distillation | 48.0 | 60.9 | 54.1 | 65.4 | 61.4 | 78.5 | 65.8 | 85.8 | 70.3 | 75.3 | 12.4 | 16.1 | 28.2 | **41.0** | 73.5 | 87.1 | 51.7 | 63.8 |
| **8** w/o CE loss on univ. | 48.0 | 60.9 | 60.5 | 70.2 | 60.5 | 77.7 | 67.2 | 86.4 | 69.6 | 74.9 | 12.6 | 15.9 | 25.8 | 37.8 | 74.5 | 87.5 | 52.3 | 64.0 |
| **9** Dyn Sampler on univ. | 48.2 | 60.8 | 60.6 | 70.8 | 64.3 | 80.4 | 69.6 | 88.3 | 70.1 | 75.2 | **13.2** | **16.5** | 27.2 | 39.6 | 75.2 | 87.7 | 53.6 | 64.9 |

Table 2: Ablation studies for the performance of the universal embedding, given different modifications of the Full UDON approach. The comparison is performed on the test set of UnED.

row 4 – compare with row 3). In both experiments, the dynamic sampler delivers a significant boost in the two most difficult (instance level) domains Met and GLDv2, similar performance in iNat, and a drop in the other 5 domains. All following ablations are performed with the dynamic sampler.

**Distillation objectives.** Two distillation losses are involved in training the full UDON method: the relational distillation loss (2) and the logit distillation loss (3). Both the losses improve the performance, as can be seen in Table 2 comparing the Full method (row 3), relational distillation only (w/o logit distillation, row 6), and no distillation loss (row 7).

**Classification loss** on the universal embedding. Removing the classification loss from the universal embedding ("w/o CE loss on univ.", row 8) results in a loss of average performance. Interestingly, it has a slightly positive impact on the Met domain.

**Online teacher dimensionality.** We perform an experiment ("64-D teachers", row 5) with dimensionality of the specific domain teachers reduced to 64D (*i.e.* the same dimensionality as of the universal student embedding) as compared to 256D in the full method. This results in a performance drop, which aligns with previous observations that higher dimensional embeddings can be better teachers in a distillation setting [36].

**Scheduling the dynamic sampler.** The sampler probability is updated every 1000 optimization steps (UDON needs ∼120k steps to converge). Our experiments show that the method is not sensitive to the choice of this parameter. The probability is updated according to the training classification loss of the current model on each domain. In fact, there are two such losses in UDON. One provided by each domain teacher's classifier, and one provided by the classification loss on the universal student's separate classifier for each domain. Using the latter to update the sampler's weights incurs a small drop in performance, as seen in ("Dyn Sampler on univ.", row 9), compared to the Full UDON method, where the domain teacher's classification loss updates the sampler.

## 4.4 Other distillation approaches

The online distillation method of UDON provides a very efficient way of transferring domain-specific knowledge to the universal embedding, without training more than a single backbone. In this section, the application of alternative distillation approaches is discussed. In particular, we implement two other approaches: first, the naive multi-teacher distillation (Figure 1 middle), with independent specialist models as teachers (8 extra backbones), where each teacher is trained in its own domain, dubbed "8 separate teachers". Second, one model with specialist heads is trained, *i.e.* 1 extra backbone followed by domain-specific projections (teacher embeddings), dubbed "1 separate teacher". In both cases, the teacher backbones are fixed during distillation, and the universal embedding (student) gets its own backbone. All backbones are initialized by ImageNet21k in the experiments of Tables 3 and 4. For efficiency reasons, only relational distillation is performed in this experiment, and all the teacher embeddings are 256 dimensional, as in UDON.

**Universal embedding performance.** The results are presented in Table 3, indicating that our method is not only efficient, but also outperforms other variants. This finding indicates that UDON benefits significantly from sharing the backbone between the student and the teachers, even if that could limit the representation capacity of the teachers, given that they have fewer free parameters.

**Compute cost reduction.** The alternative approaches are less efficient in terms of the number of parameters, as well as in the number of steps needed to converge. More specifically, for this setup, UDON uses ∼188 million (M) parameters, "1 separate teacher" uses ∼440M parameters, and "8

| | Food2k | | CARS196 | | SOP | | InShop | | iNat | | Met | | GLDv2 | | Rp2k | | Mean | |
|---|---|---|---|---|---|---|---|---|---|---|---|---|---|---|---|---|---|---|
| Model | P@5 | R@1 | P@5 | R@1 | P@5 | R@1 | P@5 | R@1 | P@5 | R@1 | P@5 | R@1 | P@5 | R@1 | P@5 | R@1 | P@5 | R@1 |
| 8 separate teachers | 47.6 | 60.8 | 58.0 | 69.0 | **62.8** | **79.4** | 67.9 | **87.3** | 70.1 | 75.2 | **12.5** | **15.8** | 26.0 | 38.4 | 74.7 | 87.6 | 52.4 | 64.2 |
| 1 separate teacher | 47.2 | 59.8 | 54.7 | 66.1 | 62.5 | 79.3 | 66.5 | 86.6 | 69.9 | 75.1 | 12.0 | 15.1 | 26.2 | 38.8 | 74.2 | 87.5 | 51.7 | 63.6 |
| **UDON (Ours)** | **49.2** | **61.8** | **60.1** | **70.4** | 62.6 | 79.2 | **68.0** | 87.2 | **70.3** | **75.3** | **12.5** | 15.6 | **28.7** | **41.0** | **74.9** | **87.8** | **53.3** | **64.8** |

Table 3: Performance comparison of the universal embedding using different distillation approaches, on the test set of UnED. "8 separate teachers" indicates the setting of 8 independent specialist models distilling to a universal model, while "1 separate teacher" indicates the setting of 1 independent model with 8 separate domain heads distilling to a universal model.

| | Food2k | | CARS196 | | SOP | | InShop | | iNat | | Met | | GLDv2 | | Rp2k | | Mean | |
|---|---|---|---|---|---|---|---|---|---|---|---|---|---|---|---|---|---|---|
| Model | P@5 | R@1 | P@5 | R@1 | P@5 | R@1 | P@5 | R@1 | P@5 | R@1 | P@5 | R@1 | P@5 | R@1 | P@5 | R@1 | P@5 | R@1 |
| **Separate Index Evaluation (Oracle)** | | | | | | | | | | | | | | | | | | |
| 8 separate teachers | 54.7 | 66.4 | 71.8 | 81.1 | 74.9 | 87.0 | 77.7 | 92.6 | 76.7 | 81.3 | 41.2 | 49.9 | 34.7 | 49.1 | 80.6 | 90.3 | 64.0 | 74.7 |
| 1 separate teacher | 52.5 | 65.3 | 57.3 | 68.0 | 71.7 | 85.1 | 71.9 | 89.8 | 76.6 | 81.2 | 35.2 | 43.4 | 31.1 | 45.0 | 79.2 | 89.8 | 59.4 | 71.0 |
| **UDON** teachers | 52.9 | 64.8 | 62.4 | 71.6 | 72.7 | 85.6 | 75.2 | 91.8 | 74.8 | 79.6 | 29.0 | 34.6 | 32.6 | 45.2 | 79.5 | 90.1 | 59.9 | 70.4 |

Table 4: Performance comparison of the **teacher** embeddings (256D) that are used by the different distillation approaches, on the test set of UnED, **but on the separate index setting**, *i.e.* each query is only compared against the index of its own domain.

separate teachers" uses ∼873M parameters, saving as much as ∼4.5 times in parameters, while improving performance. Additionally, UDON takes on average ∼120k steps to converge, "1 separate teacher" needs around ∼220k steps (sum of the 2 training phases), "8 separate teachers" ∼250k training steps (sum of the 9 training phases), cutting the number of convergence steps in half. For reference, the no-distillation baseline USCRR converges at around the same steps as UDON.

**Teachers' performance.** To gain a better insight, we also evaluate the performance of the teachers in their domains. The domain of test image is used as an oracle in these experiments in order to restrict the index to contain images of the same domain only, hence the reported numbers are **not** comparable to other reported results. Table 4 shows that the independent specialists provide the best per-domain teachers. Interestingly, although being the best performing (teachers) in their domain, the latter do not provide the best distillation outcome, which is delivered by UDON, as discussed in the previous paragraphs. We hypothesize that sharing the backbone between the teachers and the student in UDON, on the one hand limits the performance of the teachers on their individual domains, but, on the other hand, allows for more efficient distillation, as the specialist heads and the universal student operate on the same backbone embedding. Another related observation can be made from the ablation Table 2. Row 2 ("USC w/ Dyn Sampler") and row 7 ("w/o any distillation") differ in the presence of separate domain classification heads on top of the universal student backbone in UDON, without performing distillation. In the latter method, the specialist heads provide a regularization for the backbone training, which results in improved performance.

# 5 Conclusions and Limitations

**Conclusions.** In this work, a novel multi-teacher distillation approach – Universal Dynamic Online distillatioN (UDON) – is introduced to tackle the problem of learning a universal embedding. The universal embedding and the domain-specific teachers share the backbone parameters and are trained jointly, which proves to be very efficient both in time and resources. The proposed training approach is shown to deliver high efficacy distillation, in which the universal student performs even better than distilling from separate fixed teachers. The additionally proposed difficulty-based dynamic sampling results in a significant boost of performance in complex domains which are typically characterized by a large number of classes and long-tail distributions. The proposed method improves the state-of-the-art performance on the recent UnED benchmark.

**Limitations.** While UDON boosts universal embedding performance compared to the baseline method USCRR which only employs classification loss, it has 20% decrease in training throughput, given that it adds new parameters (in the teacher heads). Additionally, the proposed dynamic sampling significantly improves the performance in the difficult domains, such as Met and GLDv2, however, still at a cost of slightly decreased performance on other simpler domains.

## 6 Acknowledgements

The authors acknowledge the support of the National Recovery Plan funded project MPO 60273/24/21300/21000 CEDMO 2.0 NPO, the Czech Technical University in Prague grant No. SGS23/173/OHK3/3T/13, and the CTU institutional support (Future fund).

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

# A  Appendix

In the Appendix we present results from Tables 1 to 3 of the main paper that include the standard deviations calculated across 3 randomizations per experiment. Additionally, we present results for different backbone sizes, a series of experiments to justify the choice of the temperature value in the main paper, and a study of the architecture of the embedding projectors used in the UDON pipeline. Lastly, additional qualitative results are presented.

## A.1  Standard Deviations

In Table 5, the results from Table 1 of the main paper, along with the corresponding standard deviations are presented. Only the methods that were developed in this work are shown, namely the baselines "USC w/ Dyn Sampler", "MLP baseline w/ Dyn Sampler" and the proposed UDON method, for ImageNet pretraining, as well as the results for UDON method for CLIP pretraining. In Table 6 the results of the ablations from Table 2 of the main paper are presented, with the corresponding standard deviations. In Table 7 the results from Table 3 of the main paper showcasing the performance of the universal embedding under different distillation methods are presented, with the corresponding standard deviations.

| | Food2k | | CARS196 | | SOP | | InShop | | iNat | | Met | | GLDv2 | | Rp2k | | Mean | |
|---|---|---|---|---|---|---|---|---|---|---|---|---|---|---|---|---|---|---|
| Model | P@5 | R@1 | P@5 | R@1 | P@5 | R@1 | P@5 | R@1 | P@5 | R@1 | P@5 | R@1 | P@5 | R@1 | P@5 | R@1 | P@5 | R@1 |
| **ImageNet21k pretraining** | | | | | | | | | | | | | | | | | | |
| USC w/ Dyn Sampler | 46.2±1.1 | 59.1±1.0 | 56.3±0.7 | 67.4±0.4 | 61.1±0.7 | 78.4±0.5 | 65.9±0.7 | 86.1±0.6 | 69.6±0.2 | 74.8±0.3 | 12.0±0.3 | 15.6±0.3 | 25.9±1.5 | 37.3±3.1 | 73.2±0.4 | 86.9±0.1 | 51.3±0.5 | 63.2±0.7 |
| MLP baseline w/ Dyn Sampler | 47.5±0.5 | 60.1±0.1 | 51.3±0.8 | 63.0±0.6 | 61.8±0.2 | 78.5±0.2 | 64.1±0.3 | 85.0±0.6 | 69.7±0.2 | 74.8±0.2 | **14.3±1.0** | **17.8±1.4** | 25.8±0.7 | 36.4±1.0 | 73.3±0.4 | 86.8±0.2 | 51.0±0.4 | 62.8±0.4 |
| UDON (Ours) | **49.6±0.6** | **62.2±0.7** | **61.3±1.2** | **71.2±1.2** | **64.2±0.7** | **80.2±0.3** | **69.8±1.1** | **88.5±0.9** | **70.4±0.7** | **75.3±0.7** | 12.0±0.3 | 15.9±0.7 | **28.6±1.2** | **40.9±0.3** | **75.6±0.3** | **88.0±0.2** | **53.9±0.2** | **65.3±0.2** |
| **CLIP pretraining** | | | | | | | | | | | | | | | | | | |
| **UDON (Ours)** | 50.3±0.6 | 62.4±0.3 | 80.0±1.3 | 85.8±0.9 | 67.0±0.9 | 82.1±0.8 | 71.8±1.7 | 89.7±0.8 | 66.7±0.5 | 72.7±0.4 | 15.8±1.5 | 19.6±1.7 | 30.9±1.2 | 43.4±1.0 | 72.7±0.7 | 85.9±0.2 | 56.9±0.4 | 67.7±0.1 |

Table 5:  Evaluation results for the methods developed in this work, with the corresponding standard deviations across the 3 randomizations. This table corresponds to Table 1 of the main paper.

| | Food2k | | CARS196 | | SOP | | InShop | | iNat | | Met | | GLDv2 | | Rp2k | | Mean | |
|---|---|---|---|---|---|---|---|---|---|---|---|---|---|---|---|---|---|---|
| Model | P@5 | R@1 | P@5 | R@1 | P@5 | R@1 | P@5 | R@1 | P@5 | R@1 | P@5 | R@1 | P@5 | R@1 | P@5 | R@1 | P@5 | R@1 |
| 2 USC w/ Dyn Sampler | 46.2±1.1 | 59.1±1.0 | 56.3±0.7 | 67.4±0.4 | 61.1±0.7 | 78.4±0.5 | 65.9±0.7 | 86.1±0.6 | 69.6±0.2 | 74.8±0.3 | 12.0±0.3 | 15.6±0.3 | 25.9±1.5 | 37.3±3.1 | 73.2±0.4 | 86.9±0.1 | 51.3±0.5 | 63.2±0.7 |
| **UDON (Ours)** | | | | | | | | | | | | | | | | | | |
| 3 Full UDON | 49.6±0.6 | 62.2±0.7 | 61.3±1.2 | 71.2±1.2 | 64.2±0.7 | 80.2±0.3 | 69.8±1.1 | 88.5±0.9 | **70.4±0.7** | 75.3±0.7 | 12.0±0.3 | 15.9±0.7 | 28.6±1.2 | 40.9±0.5 | 75.6±0.3 | 88.0±0.2 | 53.9±0.2 | **65.3±0.2** |
| 4 w/o Dyn Sampler (RR) | **50.3±0.6** | **62.9±0.4** | **62.9±1.4** | **72.5±1.6** | **67.4±0.6** | **82.2±0.3** | **75.4±0.1** | **90.8±0.2** | 70.4±0.2 | **75.4±0.3** | 7.6±0.6 | 9.4±0.9 | 23.3±1.6 | 33.4±2.2 | **76.8±1.1** | **88.6±0.7** | **54.3±0.1** | 64.4±0.2 |
| 5 64-D teachers | 47.7±0.8 | 60.3±0.5 | 60.2±0.6 | 70.3±0.7 | 64.3±1.2 | 80.3±0.7 | 68.6±1.5 | 87.6±0.9 | 69.9±0.3 | 75.0±0.3 | 11.3±1.6 | 14.5±1.9 | 26.4±1.1 | 38.1±1.2 | 74.2±0.9 | 87.6±0.4 | 52.8±0.5 | 64.2±0.2 |
| 6 w/o logit distillation | 49.2±0.3 | 61.8±0.2 | 60.1±0.7 | 70.4±0.5 | 62.6±0.9 | 79.2±0.6 | 68.0±0.7 | 87.2±0.3 | 70.3±0.2 | 75.3±0.4 | 12.5±1.3 | 15.6±1.4 | **28.7±1.5** | **41.0±1.4** | 74.9±0.8 | 87.8±0.5 | 53.3±0.2 | 64.8±0.1 |
| 7 w/o any distillation | 48.0±0.6 | 60.9±0.6 | 54.1±1.0 | 65.4±0.8 | 61.4±0.5 | 78.5±0.2 | 65.8±0.3 | 85.8±0.4 | 70.3±0.1 | 75.3±0.1 | 12.4±0.9 | 16.1±0.5 | 28.2±0.4 | 41.0±0.7 | 73.5±0.2 | 87.1±0.2 | 51.7±0.2 | 63.8±0.2 |
| 8 w/o CE loss on univ. | 48.0±0.9 | 60.9±0.9 | 60.5±1.2 | 70.2±2.0 | 60.5±1.5 | 77.7±1.0 | 67.2±1.7 | 86.4±1.1 | 69.6±0.5 | 74.9±0.3 | 12.6±1.1 | 15.9±1.5 | 25.8±1.0 | 37.8±1.0 | 74.5±0.9 | 87.5±0.4 | 52.3±0.8 | 64.0±0.5 |
| 9 Dyn Sampler on univ. | 48.2±1.6 | 60.8±1.1 | 60.6±0.5 | 70.8±0.7 | 64.3±1.5 | 80.4±0.8 | 69.6±2.2 | 88.3±1.2 | 70.1±0.4 | 75.2±0.3 | **13.2±0.6** | **16.5±0.5** | 27.2±2.0 | 39.6±1.7 | 75.2±0.4 | 87.7±0.3 | 53.6±1.0 | 64.9±0.6 |

Table 6:  Evaluation results for the ablation studies from Table 2 of the main paper, along with the standard deviations calculated across the 3 randomizations of each experiment.

| | Food2k | | CARS196 | | SOP | | InShop | | iNat | | Met | | GLDv2 | | Rp2k | | Mean | |
|---|---|---|---|---|---|---|---|---|---|---|---|---|---|---|---|---|---|---|
| Model | P@5 | R@1 | P@5 | R@1 | P@5 | R@1 | P@5 | R@1 | P@5 | R@1 | P@5 | R@1 | P@5 | R@1 | P@5 | R@1 | P@5 | R@1 |
| **Distillation models** | | | | | | | | | | | | | | | | | | |
| 8 separate teachers | 47.6±0.9 | 60.8±0.9 | 58.0±0.6 | 69.0±0.6 | **62.8±1.0** | **79.4±0.6** | 67.9±1.3 | **87.3±0.7** | 70.1±0.3 | 75.2±0.2 | **12.5±0.8** | **15.8±0.9** | 26.0±1.3 | 38.4±1.7 | 74.7±0.7 | 87.6±0.6 | 52.4±0.6 | 64.2±0.5 |
| 1 separate teacher | 47.2±1.2 | 59.8±1.0 | 54.7±0.6 | 66.1±0.9 | 62.5±0.8 | 79.3±0.4 | 66.5±0.8 | 86.6±0.5 | 69.9±0.3 | 75.1±0.2 | 12.0±1.0 | 15.1±1.9 | 26.2±1.2 | 38.8±1.0 | 74.2±0.7 | 87.5±0.2 | 51.7±0.7 | 63.6±0.4 |
| UDON (Ours) | 49.2±0.3 | 61.8±0.2 | 60.1±0.7 | 70.4±0.5 | 62.6±0.9 | 79.2±0.6 | 68.0±0.7 | 87.2±0.3 | 70.3±0.2 | 75.3±0.4 | 12.5±1.3 | 15.6±1.4 | 28.7±1.5 | 41.0±1.4 | 74.9±0.8 | 87.8±0.5 | 53.3±0.2 | 64.8±0.1 |

Table 7:  Evaluation results for the comparisons between distillation alternatives from Table 3 of the main paper, along with the standard deviations calculated across the 3 randomizations of each experiment.

## A.2  UDON with a Smaller Backbone

We present additional results for the UDON method, for a smaller backbone size, namely ViT-Small/16 in Table 8, where we compare with the baseline method UJCRR of [45], which is one of the best performing methods. Both methods start from ImageNet21k pretraining. The results indicate that our method is also applicable to smaller backbone sizes.

## A.3  UDON with a Larger Backbone

In order to examine if the performance gain achieved by UDON with a ViT-Base (over the baseline USCRR) diminishes for larger backbone sizes, we performed additional experiments with the larger Vision Transformer variant, namely the ViT-Large model with ImageNet21k pre-training. The results are presented in Table 9. We observe that the USCRR baseline with a larger backbone achieves almost the same performance as its smaller ViT-Base counterpart, indicating that a larger backbone

|  | Mean | |
|---|---|---|
| Model | P@5 | R@1 |
| ViT-S (ImageNet21k) + UJCRR [45] | 48.3 | 58.9 |
| ViT-S (ImageNet21k) + **UDON (Ours)** | **49.1** | **61.1** |

Table 8: Performance comparison of the proposed UDON method with the UJCRR method of [45] on the test set of UnED, for the smaller backbone size of ViT-Small.

size doesn't necessarily mean better performance (note that a similar observation is made in [45], Appendix A, section A.2). Additionally, the UDON-trained ViT-Large achieves a performance that is better but close to the ViT-Base counterpart, indicating both the effectiveness of the UDON training procedure for the larger backbone size compared to the baseline training procedure, as well as the fact that the ViT-Base achieves a very good size-performance tradeoff.

|  | Mean | |
|---|---|---|
| Model | P@5 | R@1 |
| ViT-B + USCRR [45] | 51.4 | 62.5 |
| ViT-L + USCRR [45] | 51.0 | 62.4 |
| ViT-B + **UDON (Ours)** | 53.9 | 65.3 |
| ViT-L + **UDON (Ours)** | 54.6 | 65.4 |

Table 9: Performance comparison of the proposed UDON method and USCRR method of [45] on the test set of UnED, for two different backbone sizes, namely the ViT-Base and the larger backbone size of ViT-Large.

## A.4 Larger Backbone off-the-shelf models

We provide some additional results for the larger off-the-shelf models of CLIP and DINOv2 pretraining, which utilize the ViT-Large (ViT-L) backbone, in Table 10. Both utilize 1024D embeddings.

|  | Mean | |
|---|---|---|
| Model | P@5 | R@1 |
| **Off-the-shelf** | | |
| ViT-B CLIP [33](**768-D**) | 39.8 | 53.5 |
| ViT-L CLIP [33](**1024-D**) | 44.5 | 58.3 |
| ViT-B DINOv2 [26](**768-D**) | 43.9 | 58.2 |
| ViT-L DINOv2 [26](**1024-D**) | 46.7 | 60.8 |

Table 10: Additional results on the test set of UnED for the off-the-shelf models CLIP and DINOv2, which utilize the larger ViT-L backbone variant. ViT-B results are shown as well for comparison.

## A.5 Temperature of logit distillation

The value of the temperature hyperparameter was tuned on the validation set of UnED independently of other hyperparameters and kept fixed. We provide additional results alternating the temperature value in the full UDON method (IN21k pre-trained) in Table 11. The results indicate that the different values of (1,0.05,0.01) perform worse or cause the training to diverge, than the one used in the main paper (0.1).

## A.6 Architecture of the projectors in UDON

We performed an experiment where we replace the linear layers used as the projection for both the domain-specific teachers and the universal student by a deeper network. More specifically, we use a one hidden layer MLP with layernorm and GELU activation, with the same hidden dimension as the output embedding dimension, i.e. 256 for the teachers and 64 for the student. The obtained results shown in Table 12 (IN21k pre-trained ViT-Base) indicate a significant drop compared to using the proposed linear layers.

|       | Mean | |
| --- | --- | --- |
| $T$ | P@5 | R@1 |
| 0.1 | **53.9** | **65.3** |
| 1.0 | 53.6 | 65.0 |
| 0.05 | 53.2 | 64.8 |
| 0.01 | Diverged | |

Table 11: Study for different values of the temperature hyperparameter used in the UDON method. The results are shown on the test set of UnED.

|       | Mean | |
| --- | --- | --- |
| Model | P@5 | R@1 |
| UDON | **53.9** | **65.3** |
| UDON - MLP projectors | 51.8 | 63.5 |

Table 12: Performance comparison of the proposed UDON method to a version of it where the embedding projections are changed from linear layers to MLPs (UDON - MLP projectors), on the test set of UnED.

## A.7 Additional Qualitative Results

In Figure 4, additional qualitative results for queries that the UDON universal embedding outperforms the USCRR [45] baseline universal embedding are shown. Note that for queries whose class is represented by less than 5 positives in the index, we present as many neighbors as the number of positives, since only those are taken into account for the calculation of the metrics.

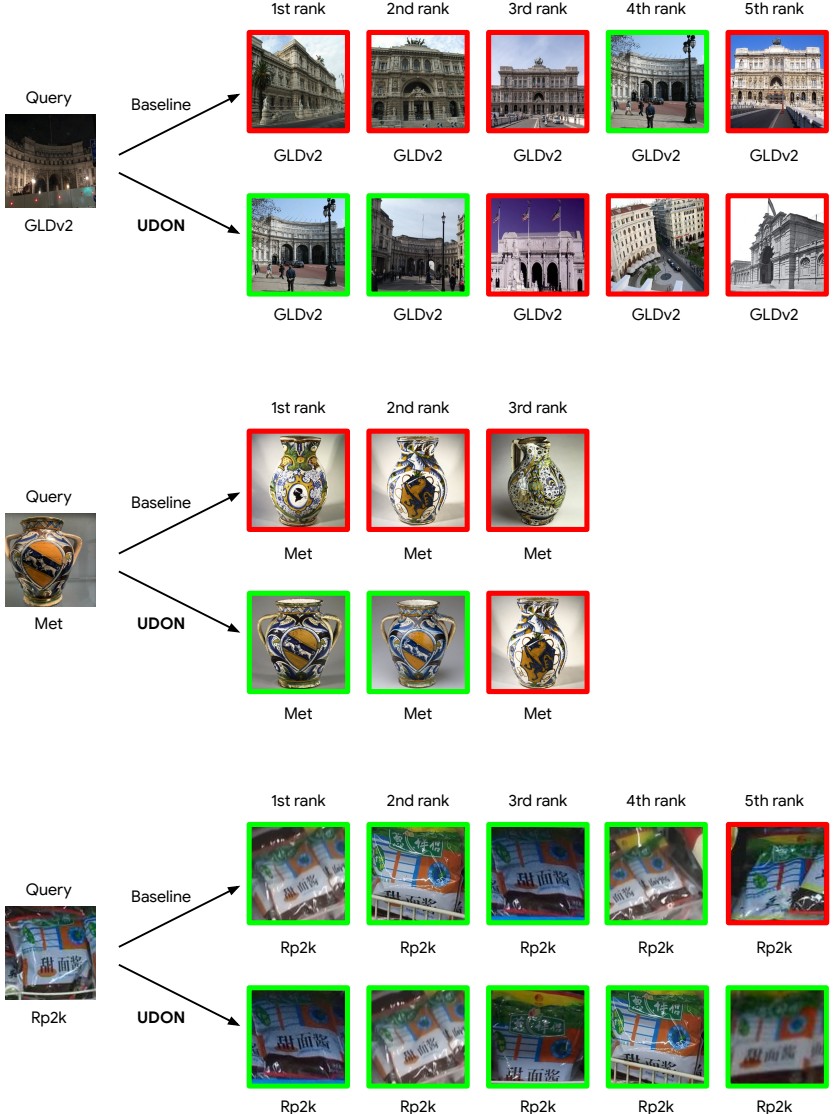

Figure 4: **Additional qualitative results for our UDON method.** We present nearest neighbours that are retrieved by the baseline (USCRR) embedding (top row) and the proposed UDON embedding (bottom row), for queries that the proposed method improves over the baseline. Each image shows the domain it comes from (underneath it). The correct neighbors are in green border, the incorrect ones are in red. For queries whose class is represented by less than 5 positives in the index, we present as many neighbors as the number of positives, since only those are taken into account for calculating the metrics.

