# OpenReview forum: "UDON: Universal Dynamic Online distillatioN for generic image representations"
_NeurIPS.cc/2024/Conference — NeurIPS 2024 poster_

### Official Review · Reviewer_uq4A · 2024-07-11

**Soundness:** 2
**Presentation:** 3
**Contribution:** 2
**Rating:** 3
**Confidence:** 4

**Summary:**

The paper introduces a novel method for enhancing universal image embeddings through a multi-teacher knowledge distillation approach. The method, named UDON, employs a dynamic sampling technique and a shared backbone across multiple domain-specific teachers and a universal student model to efficiently distill domain-specific knowledge into a universal embedding. This embedding model can work as a foundation for many downstream task models, such as classification, retrieval, generation, etc., with a high potential impact on the community.

**Strengths:**

1) The introduction of multi-teacher distillation with a shared backbone is novel and addresses significant challenges in universal image representation.

(2) The dynamic sampling method that adapts based on domain-specific performance enhances learning efficiency and addresses the imbalance in training data distribution.

(3) Extensive experiments demonstrate that UDON outperforms existing methods, showcasing the effectiveness of the proposed techniques.

**Weaknesses:**

(1) This paper misses a series of very important works for embedding learning [1,2]. The [1] proposed the Matryoshka representation learning (MRL) that the OpenAI's recent embedding model adopts. MRL would unfold the embedding model into multiple dimensions such as 64, 128, 512 or more. This paper simply takes the 64 and cannot be extended to higher dimensions which significantly limited its potential in applications.

[1] Kusupati, Aditya, et al. "Matryoshka representation learning." Advances in Neural Information Processing Systems 35 (2022): 30233-30249.

[2] Cai, Mu, et al. "Matryoshka Multimodal Models." arXiv preprint arXiv:2405.17430 (2024).

(2) Tab 1 is pretty confusing. There are "Off-the-shelf", "Specialist+Oracle", "ImageNet21k pretraining" and "CLIP pretraining". It is hard to understand the exact meaning of them. Moreover, the evaluation is not convincing. It misses some recent SOTA embedding models, such as SigLIP.

(3) Please unify the format of subtitles. In Line 253, there is "Implementation Details" but line 235 applies "Compared methods".

(4) There is no ablation study over the temperature and embedding dimension, which should be critical hyperparameters to explore.

**Questions:**

N/A

---

> ### Author Rebuttal · Authors · 2024-08-06
>
> > __This paper misses a series of very important works for embedding learning [1,2]. The [1] proposed the Matryoshka representation learning (MRL) that the OpenAI's recent embedding model adopts. MRL would unfold the embedding model into multiple dimensions such as 64, 128, 512 or more. This paper simply takes the 64 and cannot be extended to higher dimensions which significantly limited its potential in applications.__
>
> The main focus of our paper is on improving universal embedding models, following the setup proposed by [44]. Note that in the experimental setup of [44] the embedding dimension must be fixed to 64, which is the reason we chose this dimensionality. Regarding MRL, it is a technique proposed to enable adaptive embedding dimensions in deployment. We agree that this is an interesting direction, but it is not directly related to the topic explored in our paper, which proposes techniques to improve universal image embeddings in general. Combining UDON and MRL in the future would be an interesting exploration.
>
> > __Tab 1 is pretty confusing. There are "Off-the-shelf", "Specialist+Oracle", "ImageNet21k pretraining" and "CLIP pretraining". It is hard to understand the exact meaning of them.__
>
> The meaning of different settings is explained in section 4.1 "Compared methods", 4.2 and the caption of Table 1. The concept of the table is taken from [44 - Table 4].
>
> > __Moreover, the evaluation is not convincing. It misses some recent SOTA embedding models, such as SigLIP.__
>
> As mentioned in the "Compared methods" paragraph in Subsection 4.1 "Experimental settings", the Universal Embedding task and the corresponding UnED benchmark is only recently created, and we have done our best to include as many baselines as we can into the evaluation. We appreciate the reviewer additionally pointing us to the recent SigLIP model. We evaluated the SigLIP ViT-Base off-the-shelf model, which achieves a mean P@5 of 49.0% and a mean R@1 of 62.0% on the UnED benchmark. We will add it to the final version of the paper.
>
> > __Please unify the format of subtitles. In Line 253, there is "Implementation Details" but line 235 applies "Compared methods".__
>
> Thank you for this one, good catch! We will fix this by replacing "D" with "d".
>
> > __There is no ablation study over the temperature and embedding dimension, which should be critical hyperparameters to explore.__
>
> The value of the temperature hyperparameter was tuned on the validation set independently of other hyperparameters and kept fixed. For the rebuttal, we provide additional numbers alternating the temperature value in the full UDON method (IN21k pre-trained):
>
> | temperature value |mean P@5(%)|mean R@1(%)|
> |------|------|------|
> | 0.1 (paper)  |  53.9  | 65.3   |
> | 1.0    | 53.6  |  65.0  |
> | 0.05  | 53.2  | 64.8 |
> | 0.01  | - (Diverged)  | - (Diverged)  |
>
> all of which perform significantly worse than the one used in the paper. We will add this study to the final version of the paper.
>
> For the teacher embedding dimension, there is already an ablation in the paper, see Table 2 row 5, and corresponding paragraph "Online teacher dimensionality" in Subsection 4.3.
> Regarding the student embedding dimensionality, we keep it fixed to 64, given the constraints of the UnED benchmark [44], as discussed above.

---

> ### Comment · Reviewer_uq4A · 2024-08-10
> **After Rebuttal**
>
> Thank you for the detailed response from the authors. I will maintain my original score, as the novelty and impact of the work seem limited. The concept of multi-teacher distillation is not new, and testing solely on UnED with fixed dim size may not be sufficiently convincing.

---

### Official Review · Reviewer_xwWg · 2024-07-12

**Soundness:** 3
**Presentation:** 2
**Contribution:** 3
**Rating:** 6
**Confidence:** 3

**Summary:**

The paper proposes Universal Dynamic Online distillatioN (UDON), which is a multi-teacher distillation method designed for universal image representations. UDON adopts a knowledge distillation strategy by distilling information from multiple teacher model trained for different domains to a student model to learn the universal embedding. It also proposes a dynamic domain sampling strategy for balancing the different domains. The provided experimental results verify its effectiveness.

**Strengths:**

The proposed design is simple and elegant. The authors describe the design and implementation in details, which makes it easy to follow.

The experimental results also demonstrate its effectiveness. In addition, many ablation experiments are conducted to give insight.

I suppose the paper is valuable enough to be accepted.

**Weaknesses:**

**The reason why the proposed method is better than the previous work (USC) is not straightforward and clear enough for me.**
In the paper, the main baseline is Universal Separate Classifier Training method (USC). It uses a backbone to learn an universe embedding where multiple classifier heads for different domains are trained separately with the universe embedding as input. Compared with USC, UDON introduces distillation and dynamic domain sampling. I understand the part about dynamic domain sampling but feel confused about the distillation. As shown in Figure 2, UDON uses multiple extra classifier heads as teachers for distillation. Since the teacher heads and student head are all based on the previous embedding $E_b$, why the distillation strategy can help to build an better universal embedding $E_u$?

The authors explain the reason mainly on [Line 37 - Line 51] and [Line 164 - Line 169], while I am still confused about it.
On [Line 37 - Line 51], it says that "it is difficult to encode detailed knowledge about many image domains in a single model". Therefore, the authors propose to use knowledge distillation between a student and teachers model for different domains, while the student and teachers share the backbone. Does it mean that the difficulty remains for USC, but it is solved by UDON with the distillation? The story seems a bit conflicting to me. Because from my point of view, a backbone with more classifier heads is still a single model, since the heads are mostly very light-weight compared with the backbone.

In my opinion, the proposed distillation design is filtering the information from $E_b$, which keeps more useful and universal information for the student head to learn $E_u$.

**Questions:**

See weakness.

**Limitations:**

See weakness.

---

> ### Author Rebuttal · Authors · 2024-08-06
>
> We thank the reviewer for the comment about how the UDON method works compared to the USC baseline. We will provide some clarifications and explain our interpretation of the key components that allow UDON to produce a better universal embedding than previous work, based on the findings from the experiments of the paper.
>
> The extra classifier heads used in UDON as teachers, mentioned by the reviewer, each consist of a shallow domain-specific projection (which produces the domain-specific embeddings) plus a linear classifier. They stem from the same backbone embedding $E_b$ as the universal embedding $E_u$ does, and they are not necessarily lightweight (given the high dimensionality of the domain-specific embeddings + the large number of classes in the different subdomains). We will refer to them as domain-specific heads from now on.
>
> Building from USC + Dyn. Sampler for a fair comparison, appending these domain-specific heads on the side of the universal embedding that share the same backbone and training them with classification training loss from domain-specific data already results in a performance boost for the universal embedding; see the experiment in Ablation studies Table 2, rows 7 and 2. We hypothesize that these extra domain heads backpropagating through the shared backbone act as a form of regularization.
>
> On top of that, adding the distillation losses to obtain the full UDON method results in a large performance boost (Table 2, rows 7 and 3 of the table). We hypothesize that the domain-specific embeddings have additionally captured features that the universal embedding would otherwise skip. These features are used by UDON as an extra supervisory signal (teachers) to the universal embedding in the form of similarities between embeddings and learnable class prototypes.
>
> In UDON, not allowing the teachers to utilize their own backbone is not only efficient in terms of compute, but allows for a higher-performing distillation process. Our experiments show that the UDON teachers (the domain-specific embeddings) underperform domain-specific embeddings that utilize their own backbone (8 separate fixed teachers case) when evaluated on their own domain (see Table 4 row 1 vs 3). However, the resulting student universal embedding of UDON performs better than the corresponding universal embedding produced by the distillation from 8 separate fixed teachers (see Table 3 row 1 vs 3, subsection 4.4). We hypothesize that distilling from the UDON teachers is an easier task than distilling from different networks that utilize their own backbone, as the UDON teachers stem from the same embedding and are only a shallow transformation of it, so the domain-specific spaces they capture are more compatible with each other.

---

> > ### Comment · Reviewer_xwWg · 2024-08-14
> >
> > Thanks for your detailed response. After reading the response and other reviewers' comments, I will keep my original score.

---

### Official Review · Reviewer_CCei · 2024-07-13

**Soundness:** 3
**Presentation:** 4
**Contribution:** 3
**Rating:** 6
**Confidence:** 4

**Summary:**

The paper tackles the problem of multi-domain fine-grained instance recognition/retrieval. The authors propose to train a unified backbone for all modalities with online distillation with domain-specific teachers, improving the performance compared to naive single backbone baselines and being competitive with expensive methods that utilize a number of domain-specific specialists. The authors propose a number of technical contributions in addition to the online distillation, including dynamic resampling based on a proxy of the task difficulty. UDON exhibits favorable performance compared to a number of baselines on the UnED benchmark consisting of a high number of fine-grained domain-specific benchmarks.

**Strengths:**

- The paper is very well written with clarity and sufficient details about the implementations and the intuitive explanation of the main contributions.
- UDON exhibits improvements over strong baselines for a variety of tasks.
- The proposed dynamic domain sampling is interesting and achieves a clear boost to the performance for challenging benchmarks like Met and GLDv2.
 - The paper includes many ablations about the different design choices which helps in understanding where the gains stem from.

**Weaknesses:**

- [important] The paper tackles generalization for instance recognition/retrieval systems, but the scaling axis of this question has not been studied. This is lacking since in the past few years we have witnessed scalable pre-training in terms of data and parameters being a very effective solution to many generalization problems.
- The performance gains provided by UDON seem highly sensitive to which dataset is used for the evaluation. For example, while the dynamic sampling contribution helps two datasets, it dropped the performance of the other five. This does not mean that this contribution should be dismissed but rather we might need more work to achieve its objectives without hurting the performance of other stable benchmarks.
- Some ablations of the projections architecture as well as the design of the MLP baseline would be a welcome addition.

**Questions:**

1) depending on the number of classes the magnitude of the loss can be very different. How is this handled in equation [5]?
2) an important question is whether scaling is indeed the answer for most problems. For the off-the-shelf baselines, it can be easy to test much larger backbones (e.g. DinoV2-G, Clip-Large, …). It is interesting to see if simply scaling general-purpose pre-training would be sufficient to address the issues tackled in the paper.
3) Related to the question above, given that the performance gain between a naive unified model vs UDON is significant but not enormous, I wonder if scaling the naive single model baseline’s capacity slightly would be sufficient to address the shortcomings of training one model with many fine-grained domains.

**Limitations:**

Yes

---

> ### Author Rebuttal · Authors · 2024-08-06
>
> > __[important] The paper tackles generalization for instance recognition/retrieval systems, but the scaling axis of this question has not been studied. This is lacking since in the past few years we have witnessed scalable pre-training in terms of data and parameters being a very effective solution to many generalization problems. ... an important question is whether scaling is indeed the answer for most problems. For the off-the-shelf baselines, it can be easy to test much larger backbones (e.g. DinoV2-G, Clip-Large, …). It is interesting to see if simply scaling general-purpose pre-training would be sufficient to address the issues tackled in the paper.__
>
> We examined larger variants of the off-the-shelf models used in this work, like the CLIP ViT-Large and the DINOv2 ViT-Large, both using a 1024 dimensional embedding. We present their results below:
>
> | off-the-shelf model|mean P@5(%)| mean R@1(%)|
> |--|--|--|
> | CLIP ViT-Base (768-D)    |39.8|53.5|
> | CLIP ViT-Large (1024-D)    |44.5| 58.3|
> | DINOv2 ViT-Base (768-D)  |43.9| 58.2|
> | DINOv2 ViT-Large (1024-D)  |46.7|60.8|
> | UDON ViT-Base (64-D, CLIP pretrained)|56.9|67.7|
>
> Those larger models are still underperforming the smaller ViT-Base UDON model with 64-dimensional embedding.
> This result shows that simply scaling the general-purpose foundational models is not sufficient to address the issues tackled in the paper.
> We will add the corresponding results and discussion in the final version of the paper.
>
> > __Related to the question above, given that the performance gain between a naive unified model vs UDON is significant but not enormous, I wonder if scaling the naive single model baseline’s capacity slightly would be sufficient to address the shortcomings of training one model with many fine-grained domains.__
>
> This is a valid point raised by the reviewer.
> In order to examine if the performance gain achieved by UDON with a ViT-Base (over the baseline USCRR) diminishes if we scale the backbone size, we performed additional experiment with the larger backbone size of ViT-Large (IN21k pre-trained).
> The results are shown in the following table:
>
> |model| mean P@5(%)| mean R@1(%)|
> |----|-----|-----|
> | USCRR (baseline) ViT-Base |51.4|62.5|
> | USCRR (baseline) ViT-Large |51.0|62.4|
> | UDON ViT-Base|53.9|65.3|
> | UDON ViT-Large|54.6| 65.4|
>
> We observe that the baseline with larger backbone achieves almost the same performance as its smaller ViT-Base counterpart, indicating that larger backbone size doesn't necessarily mean better performance (note that a similar observation is made in [44] Appendix A, section A.2). Additionally, the UDON trained ViT-Large achieves a performance which is a bit better but close to the ViT-Base counterpart, indicating both the effectiveness of the UDON training procedure for the larger backbone size compared to the baseline training procedure, as well as the fact that the ViT-Base achieves a very good size-performance tradeoff.
> We will include this discussion in the final version of the paper.
>
> > __The performance gains provided by UDON seem highly sensitive to which dataset is used for the evaluation. For example, while the dynamic sampling contribution helps two datasets, it dropped the performance of the other five. This does not mean that this contribution should be dismissed but rather we might need more work to achieve its objectives without hurting the performance of other stable benchmarks.__
>
> Assume one (meaningful) sampling strategy, say Round Robin, which oversamples some domains and undersamples others.
> If with a new strategy some domain is sampled more than it was before, the model is expected to perform better on that domain, and the other way around. The proposed Dynamic Sampling is not perfect, but it improves the difficult domains where previous work was performing poorly and leaves the others close to the original perfomance, and most importantly improves the average performance, moving closer towards the notion of a "Universal Image Embedding", which works well on all of the different subdomains.
>
> > __depending on the number of classes the magnitude of the loss can be very different. How is this handled in equation [5]?__
>
> The Dynamic Sampling is designed to sample the difficult domains more often. There is no explicit guarantee that domains with more classes must have higher learning error. Large number of well separable classes may have smaller error that a few hard-to-distinguish classes. In general, domains with higher number of classes have higher chance to contain difficult sets of classes - and thus should be sampled more often. Therefore, the number of classes is not explicitly accounted for in equation [5].
> Additionally, a number of sampling strategies and variations on the dynamic sampling have been tried, all yielding similar or worse performance on the validation set.
> At the moment, the Dynamic sampling operates on the level of domains. Ideally, one would like to analyze the training data and sample difficult classes (and their confusers) more often, which might be time demanding. An efficient approach for such sampling is left for future work.
>
> > __Some ablations of the projections architecture as well as the design of the MLP baseline would be a welcome addition.__
>
> We performed an experiment where we replace the linear layers used as the projection for both the domain-specific teachers and the universal student by a deeper network. More specifically, we use a one hidden layer MLP with layernorm and GELU activation (standard in the literature), with the same hidden dimension as the final dimension, i.e. 256 for the teachers and 64 for the student. The obtained results are shown below (IN21k pre-trained ViT-Base):
>
> | model | mean P@5(%) | mean R@1(%) |
> |---------|------|-----|
> | UDON |53.9|65.3|
> | UDON MLP projectors|51.8|63.5|
>
> The results indicate a significant drop compared to using the proposed linear layers. We plan to include this study in the final version of the paper.

---

> > ### Comment · Reviewer_CCei · 2024-08-13
> >
> > I would like to thank the authors for the rebuttal, it has addressed the majority of my questions/concerns. Therefore, I will raise my score to 6.

---

### Official Review · Reviewer_Mmnc · 2024-07-17

**Soundness:** 3
**Presentation:** 3
**Contribution:** 3
**Rating:** 5
**Confidence:** 3

**Summary:**

The paper proposes an online distillation approach in a multi-teacher setup w/ weight sharing for efficiency. A strategic dynamic batch sampling process has been proposed to help domains w/ slower learning during training.

**Strengths:**

The unified backbone and batch sampling strategy is novel and powerful. The paper is well written and experiments/datasets carefully chosen.

**Weaknesses:**

Did not spot any, though, it seems there is a throughput drop.

**Questions:**

Can you elaborate more on impact of the throughput drop in real-world industry applications? Also, specify how practical applications could benefit - please provide specific details.

**Limitations:**

Mentioned in paper

---

> ### Author Rebuttal · Authors · 2024-08-06
>
> > __Can you elaborate more on impact of the throughput drop in real-world industry applications?__
>
> The 20% throughput drop only impacts the model training stage. The inference time of our model is exactly the same as previous state-of-the-art models from [44], which makes our inference-time comparisons fair. For industry applications, generally the model will be trained during a development cycle when compute resources are used intensively one-off to produce a high-quality model with acceptable cost. Once the development cycle ends, the finalized model will be used for inference in production for a long period of time. For this reason, the inference time cost is the main concern when developing industry models, and a 20% training cost increase is not usually a problem, given the significant quality gains.
>
> > __Also, specify how practical applications could benefit - please provide specific details.__
>
> Several practical applications can directly benefit from an improved universal image embedding. In particular, generic visual recognition systems such as [46, 1, 2] need to handle queries depicting any type of object or scene. As per news reports, these systems today receive 10B+ searches per month and their popularity continues growing. This calls for a universal embedding, since it is not scalable to handle images of different domains with specialized, per-domain models. Besides, these embeddings can be useful in many other applications. For example, retrieval-augmented generation with large language models often requires access to specific visual information from an external database, which can searched with a universal image embedding. With the ever-growing number of images in many aspects of modern life, such embeddings also become critical for searching large private photo collections or street-level photos at planet-scale, for example.

---

### Author Rebuttal · Authors · 2024-08-06

We thank the reviewers for their constructive feedback.
We are encouraged that they recognize our contributions as novel (**Mmnc**, **uq4A**), interesting (**CCei**), and simple/elegant (**xwWg**). Reviewers also highlight the value of our experimental validation/ablations (**Mmnc**, **CCei**, **xwWg**, **uq4A**), demonstrating the method’s effectiveness/improvements against previous work (**CCei**, **xwWg**, **uq4A**).
We are also glad they found the paper well-written (**Mmnc**, **CCei**), making it easy to follow (**xwWg**).
We address concerns and comments in individual responses to each reviewer below.

---

### Decision · Program_Chairs · 2024-09-25

**Decision:**

Accept (poster)

**Comment:**

This submission proposes an online distillation method to learn universal representations for multi-domain fine-grained image retrieval. The proposed method uses a shared vision transformer backbone to extract feature embeddings over images in different domains, then adds separate linear projection heads on each individual domain to construct a set of domain-specific teachers, which is used to guide the learning of a universal student representation (with another linear projection head) facilitated by a simple dynamic domain sampling strategy. The submission was finally scored (5,6,6,3) by three knowledgeable reviewers and another reviewer (scored 5 but with short and uninformative comments), who mostly acknowledged the novelty and the performance of the proposed method, and also raised some concerns about **1)** more comprehensive experiments, especially to consider data/model size scaling; **2)** more thorough experimental comparison with existing embedding learning methods; **3)** missing some ablations regarding the effect of different projector architectures in the head and different projected feature dimensions;  **4)** in-depth explanation of why multi-head domain-specific teachers are helpful to improve single-head multi-domain student, under the context of sharing a vision transformer backbone; **5)** in-depth analysis of why the proposed method is superior to counterpart methods.

The authors provided detailed responses to these concerns. Generally, two positive reviewers recognized that most of their major concerns are addressed. The negative reviewer uq4A acknowledged the authors' rebuttal, but criticized the paper's novelty and performance in final judgement, constituting a contradiction compared to the original judgement in which these points were commented as the strengths. The AC read the paper, the reviews, the rebuttal and the reviewers' feedback, and mostly agree with two positive reviewers' assessment. Therefore, I recommend to accept (poster), also considering that the novelty of the proposed method is not very strong (in online knowledge distillation research, the basic concept of using the shared network backbone and multi-head branches is not new, e.g., ONE in NeurIPS 2018 and many of its variants). The authors are encouraged to carefully consider the reviewers' comments/suggestions and their rebuttal in the final paper revision.